# Design, Synthesis and Structure-Activity Relationship of Novel Pinacolone Sulfonamide Derivatives against *Botrytis cinerea* as Potent Antifungal Agents

**DOI:** 10.3390/molecules27175468

**Published:** 2022-08-25

**Authors:** Chaojie Liu, Xiaofang Xiang, Ying Wan, Jia Yang, Yufei Li, Xinchen Zhang, Zhiqiu Qi, Lu He, Wei Liu, Xinghai Li

**Affiliations:** Department of Pesticide Science, Plant Protection College, Shenyang Agricultural University, Shenyang 110866, China

**Keywords:** chesulfamide, pinacolone, sulfonamide, synthesis, *Botrytis cinerea*

## Abstract

To develop new fungicides with high efficiency, 46 novel sulfonamide derivatives were designed and synthesized by introducing pinacolone fragment into chesulfamide which was used as lead compound. All compounds were characterized by ^1^H NMR, ^13^C NMR, and MS spectra, and the structure of compound **P-27** was also confirmed by X-ray single crystal diffraction. It was found that a variety of compounds present excellent inhibitory effect against *Botrytis cinerea*. The inhibition rates of **P-29** on tomato and strawberry were 90.24% (200 mg/L) and 100% (400 mg/L) *in vivo* respectively, which were better than the lead compound chesulfamide (59.23% on tomato seedlings and 29.63% on strawberries).

## 1. Introduction

*Botrytis cinerea* is a serious and widespread plant disease in tomato, cucumber, eggplant, pepper, grape and strawberry, especially under the condition of protected cultivation [1]. The international fungicide resistance action committee (FRAC) classified the fungi as a pathogen at high risk of fungicide resistance. The resistance of *Botrytis cinerea* to fungicides is very serious all over the world, which leads to a reduction or even failure of the control effect of fungicides [2,3,4,5]. The development of fungicides with novel action mechanism is the key to control the resistance of fungicides [6,7,8]. Chesulfamide is a novel candidate fungicide with excellent control effect on *Botrytis cinerea* without interactive resistance to multiple commercial fungicides, such as carbendazim, diethofencarb, iprodione, and procymidone, showing that it has unique mechanism of action [9,10,11]. In our previous work, several novel compounds were discovered by replacing the cyclohexanone moiety in chesulfamide with benzocyclohexanone, cyclohexanone, cyclopropyl methyl ketone, and acetophenone, respectively, which all presented superb control effect against *Botrytis cinerea* [12,13,14] (Figure 1).

Pinacolone is an intermediate widely used in the synthesis of pesticides [15,16,17]. At present, more than 20 kinds of pesticides containing pinacolone fragment have been developed, among which triazole fungicides such as *triazole*, *triadimefon*, and *paclobutrazol* are the most widely used [18,19,20] (Figure 2). Hence, the design and synthesis of novel compounds containing the structure of pinacolone still has high value and broad prospect in the development of novel pesticide. In this work, chesulfamide was used as the lead compound, and novel sulfonamide derivatives were designed and prepared by replacing the cyclohexanone moiety with pinacolone group (Figure 3). The antifungal activity against *Botrytis cinerea* was investigated by the mycelium growth method and living pot method.

## 2. Results and Discussion

### 2.1. Chemistry

The synthetic route of the target compounds is shown in Figure 1. Intermediate I was prepared from pinacolone by sulfonation with a sulfur trioxide-dioxane adduct and neutralization with potassium carbonate. Then, intermediate II was prepared by chlorination with oxalyl chloride. Finally, the target compounds were obtained by an amidation reaction with amines containing various substituents [21]. According to this synthetic method, 46 novel pinacolone sulfnamide derivatives were synthesized and their structures were characterized by ^1^H NMR, ^13^C NMR, and MS spectra (Appendix A). Additionally, the structure of compound **P-27** was also confirmed by X-ray single crystal diffraction (Figure 4).

### 2.2. Biological Assay

Table 1 summarizes the result of the in vitro antifungal spectrum of **P-1~P-46**, at a dosage of 50 mg/L against *Rhizoctonia solani*, *Fusarium graminearum*, *Sclerotinia sclerotiorum*, and *Botrytis cinerea*, which clearly shows that this series of compounds not only has good control effect on *Botrytis cinerea*, but also has good control effects on other pathogens. The inhibition rates of **P-23** and **P-30** against *Botrytis cinerea* were 86.44% and 93.35% (EC_50_ were 11.57 mg/L and 4.68 mg/L, Table 2), which were better than those of the lead compound chesulfamide.

**Table 1 molecules-27-05468-t001:** In vitro fungicidal activity of target compounds at 50 mg/L.

Compound	R	Inhibition rate(%)
*R. solani*	*F. graminearum*	*S. sclerotiorum*	*B. cinerea*
**P-1**	CF_3_CH_2_	5.33 ± 5.31	4.17 ± 8.62	9.58 ± 0	0
**P-2**	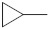	0	33.09 ± 1.06	0 ± 25.8	0
**P-3**	CH_3_(CH_2_)_3_-	19.8 ± 2.21	33.82 ± 9.46	0	0
**P-4**	CH_3_(CH_2_)_4_-	26.65 ± 1.98	21.81 ± 4.09	25.83 ± 0	42.55 ± 2.17
**P-5**	CH_3_(CH_2_)_5_-	0	10.78 ± 2.76	0 ± 3.08	19.15 ± 1.59
**P-6**	2-F-C_6_H_4_-	37.31 ± 2.68	48.04 ± 0.24	36.04 ± 2.31	25.53 ± 1.06
**P-7**	3-F-C_6_H_4_-	6.09 ± 2.86	9.56 ± 6.28	0	6.38 ± 2.31
**P-8**	4-F-C_6_H_4_-	12.94 ± 0.91	0.98 ± 4.65	32.29 ± 5.84	8.51 ± 1.74
**P-9**	2-Cl-C_6_H_4_-	20.3 ± 0.91	11.76 ± 4.24	0	22.87 ± 1.74
**P-10**	3-Cl-C_6_H_4_-	39.34 ± 7.25	22.06 ± 2.20	77.92 ± 1.81	32.45 ± 1.74
**P-11**	4-Cl-C_6_H_4_-	36.29 ± 4.08	23.53 ± 0	0	25 ± 0.46
**P-12**	2-Br-C_6_H_4_-	19.8 ± 2.92	23.77 ± 0.98	19.58 ± 0.75	26.6 ± 0.46
**P-13**	3-Br-C_6_H_4_-	9.64 ± 1.83	36.76 ± 0.42	83.75 ± 0.95	34.04 ± 4.12
**P-14**	4-Br-C_6_H_4_-	25.13 ± 0.50	36.03 ± 0.73	82.5 ± 3.81	36.17 ± 0
**P-15**	2-OCH_3_-C_6_H_4_-	9.14 ± 8.12	4.41 ± 2.94	10.42 ± 4.16	0 ± 1.32
**P-16**	2-CF_3_-C_6_H_4_-	33.76 ± 1.01	32.6 ± 2.13	55.83 ± 2.52	41.76 ± 1.15
**P-17**	3-CF_3_-C_6_H_4_-	42.89 ± 5.76	32.6 ± 0.24	36.67 ± 0	23.14 ± 0.46
**P-18**	4-OCF_3_-C_6_H_4_-	49.49 ± 3.24	4.66 ± 0.24	87.71 ± 1.10	39.1 ± 0
**P-19**	4-Cl-2-F-C_6_H_3_-	47.72 ± 2.67	45.1 ± 1.06	22.5 ± 1.80	34.04 ± 4.79
**P-20**	4-Br-2-F-C_6_H_3_-	2.54 ± 1.66	2.7 ± 2.33	0 ± 1.62	47.34 ± 0.46
**P-21**	5-CF_3_-2-F-C_6_H_3_-	37.31 ± 1.01	40.93 ± 2.09	90.42 ± 0.41	36.97 ± 5.31
**P-22**	4-Br-3-F-C_6_H_3_-	42.13 ± 8.64	46.57 ± 1.60	94.38 ± 0.55	75 ± 2.81
**P-23**	2-CF_3_-4-Cl-C_6_H_3_-	75.13 ± 2.16	69.61 ± 1.06	94.17 ± 0.62	86.44 ± 2.31
**P-24**	5-CF_3_-2-Cl-C_6_H_3_-	57.87 ± 0.43	71.57 ± 2.78	94.58 ± 1.10	64.1 ± 3.06
**P-25**	4-Br-2-NO_2_-C_6_H_3_-	53.81 ± 16.5	54.66 ± 0.24	86.04 ± 9.58	78.99 ± 0.53
**P-26**	3-Br-4-F-C_6_H_3_-	19.8 ± 1.58	5.39 ± 0.42	87.5 ± 0	27.66 ± 1.61
**P-27**	4-Br-3-CH_3_-C_6_H_3_-	34.77 ± 1.77	45.34 ± 2.48	80 ± 2.31	54.52 ± 5.07
**P-28**	2,4,5-F_3_-C_6_H_2_-	36.8 ± 3.31	30.39 ± 2.76	36.88 ± 0.75	22.07 ± 0.53
**P-29**	2,4,5-Cl_3_-C_6_H_2_-	36.8 ± 1.75	47.79 ± 0.49	80.83 ± 12.2	53.46 ± 0.79
**P-30**	2,4,6-Br_3_-C_6_H_2_-	81.47 ± 2.67	62.01 ± 4.49	97.92 ± 3.47	93.35 ± 2.17
**P-31**	C_6_H_5_-CH_2_CH_2_-	0	16.18 ± 1.36	6.25 ± 0.41	39.63 ± 0.26
**P-32**	4-NO_2_-C_6_H_4_-CH_2_CH_2_-	7.29 ± 3.51	37.33 ± 4.73	38.62 ± 6.43	33.33 ± 1.15
**P-33**	4-CH_3_-C_6_H_4_-CH_2_CH_2_-	22.92 ± 17.10	22.67 ± 5.13	51.32 ± 7.09	14.18 ± 2.52
**P-34**	4-OCH_3_-C_6_H_4_-CH_2_CH_2_-	40.63 ± 2.65	6.67 ± 4.04	42.33 ± 5.51	12.77 ± 4.36
**P-35**	3-OCH_3_-C_6_H_4_-CH_2_CH_2_-	14.58 ± 2.08	15.33 ± 3.51	38.62 ± 6.66	19.15 ± 2.65
**P-36**	3,4-(OCH_3_)_2_-C_6_H_3_-CH_2_CH_2_-	0	4.67 ± 2.52	32.28 ± 0.58	5.67 ± 4.62
**P-37**	4-F-C_6_H_4_-CH_2_CH_2_-	12.50 ± 4.00	5.33 ± 4.16	40.21 ± 6.51	21.28 ± 0
**P-38**	3-F-C_6_H_4_-CH_2_CH_2_-	10.94 ± 2.00	7.33 ± 3.79	29.63 ± 5.77	12.77 ± 5.29
**P-39**	2-F-C_6_H_4_-CH_2_CH_2_-	9.38 ± 2.65	10.67 ± 2.31	40.74 ± 4.73	15.60 ± 2.08
**P-40**	3-Br-C_6_H_4_-CH_2_CH_2_-	44.79 ± 3.79	32.00 ± 1.73	31.75 ± 1.00	25.53 ± 0
**P-41**	2-Br-C_6_H_4_-CH_2_-	46.88 ± 1.00	16.67 ± 0.58	67.20 ± 3.21	12.77 ± 3.61
**P-42**	4-F-C_6_H_4_-CH_2_-	25.52 ± 2.08	15.33 ± 3.21	23.28 ± 6.43	15.60 ± 2.08
**P-43**	2-F-C_6_H_4_-CH_2_-	25.52 ± 2.08	15.33 ± 3.21	23.28 ± 6.43	9.93 ± 1.53
**P-44**	4-Cl-C_6_H_4_-CH_2_-	16.15 ± 1.15	27.33 ± 4.93	21.69 ± 5.03	27.66 ± 4.36
**P-45**	4-CN-C_6_H_4_-CH_2_-	44.79 ± 16.86	21.33 ± 1.15	60.32 ± 6.25	17.02 ± 5.20
**P-46**	2,5-(OCH_3_)-C_6_H_4_-CH_2_-	54.69 ± 0	23.33 ± 5.51	31.75 ± 0	24.11 ± 4.62
Chesulfamide	/	80.18 ± 0.29	70.21 ± 1.28	80.18 ± 0.29	74.76 ± 4.61
Boscalid	/	85.28 ± 0.50	51.23 ± 2.72	98.33 ± 0.83	98.28 ± 0

In the tomato pot experiment, it was found that the control effect of various compounds on *Botrytis cinerea* at 200 mg/L was better than that of chesulfamide, of which compound **P-13** and **P-29** were slightly weaker than the positive control boscalid, which were 91.22% and 90.04%, respectively (Table 3, Figure 5). In the strawberry experiment, compounds **P-18**, **P-29**, **P-30** and **P-31** exhibited higher antifungal activity against *Botrytis cinerea* than chesulfamide, especially compound **P-29**, which had a 100% inhibition rate at 400 mg/L (Table 4, Figure 6).

### 2.3. Structure–Activity Relationship

Based on the result of antifungal activity experiments, the structure-activity relationship of target compounds can be concluded as follow. The order of substituent structures conducive to improving the activity of the compounds was phenyl (2,4,6-tribromophenyl is the best) > benzyl > phenylethyl > alkyl, which showed that phenyl can significantly improve the activity of the compound; and trisubstituted phenyl > disubstituted phenyl > monosubstituted phenyl. In the *in vivo* activity assay, compounds **P-18**, **P-29** and **P-31** had good antifungal activity, indicating that 4-trifluoromethoxyphenyl, 2,4,5-trichlorophenyl and phenylethyl had a positive effect on the activity of the compounds. The substituent of **P-29** is 2,4,5-trichlorophenyl, which has been found to have a good control effect on *Botrytis cinerea* in previous studies [22,23]. Therefore, this substituent can be used as the active group for structure design in the future.

## 3. Materials and Methods

### 3.1. Instruments and Reagents

For all reactions, solvents and chemical reagents were purchased from commercial sources and used as received. All reactions were carried out under a nitrogen atmosphere unless noted. Reactions were monitored by thin layer chromatography (TLC) visualizing with ultraviolet light (UV) and phosphomolybdic acid (PMA) stain. Column chromatography purification was performed using silica gel. Proton nuclear magnetic resonance (^1^H NMR) spectra and carbon nuclear magnetic resonance (^13^C NMR) were recorded on a 600 MHz spectrometer in CDCl_3_ or DMSO-*d*_6_ as the solvent and TMS as the internal standard. NMR data were presented as follows: chemical shift (δ ppm), multiplicity (s = singlet, d = doublet, t = triplet, q = quartet, m = multiplet, br = broad), coupling constant in Hertz (Hz), integration. High-resolution mass spectrometry (HRMS) data were obtained on an Agilent 1290-6540B Q-TOF instrument. Single crystal structure analysis was performed using X-ray diffraction on a Rigaku Oxford Diffraction Supernova Dual Source diffractometer.

### 3.2. General Synthetic Procedures

*Synthesis of intermediate I*. A mixture of pinacolone (1 mol) in DCE (300 mL) was stirred in a dry flask under nitrogen atmosphere at 0 °C. Then, the trioxide-dioxane adduct (1 mol) was slowly added into the flask, and the reaction mixture was stirred for 3 h. CaCO_3_ was slowly added to adjust the pH of the reaction mixture to 7. The mixture was then filtered and the pH of the mixture was adjusted to 9 by K_2_CO_3_, then filtered. The aqueous layer was collected and distilled to give the intermediate I.

*Synthesis of intermediate II.* Under the condition of ice bath, oxalyl chloride (20.6 mmol) was slowly added into the mixture of intermediate I (13.7 mmol), CH_2_Cl_2_ (40 mL) and DMF (0.54 mL), and the mixture was stirred for 3 h. Finally, a yellow liquid as obtained by suction filtration.

*Synthesis of target compounds*. A mixture of amine (9.6 mmol), CH_2_Cl_2_ (40 mL), Et_3_N (1.2 mL) was stirred in a dry flask in an ice-bath. Then, intermediate II (9.6 mmol) was added dropwise and the mixture was reacted for 3h. Upon completion, the mixture was filtered and extracted with DCM, washed with brine, dried over Na_2_SO_4_, concentrated in vacuo and purified by silica column chromatography (PE:EA = 10:1, *v*/*v* ) to give target compounds.

The spectral data of the Compound P-1~P-46 are described below.

*Data for 3,3-dimethyl-2-oxo-N-(3,3,3-trifluoroethyl)butane-1-sulfonamide* (**P-1**): Yield 38%; white solid; m.p. 121.5–123.1 °C. ^1^H-NMR (600 MHz, DMSO-*d*_6_) δ 8.15 (t, *J* = 6.7 Hz, 1H, SO_2_-*N*H)), 4.62–4.58 (m, 2H, SO_2_-CH_2_), 3.79 (qd, *J* = 9.4, 6.5 Hz, 2H, *N*-CH_2_), 1.11 (d, *J* = 0.9 Hz, 9H, CH_3_).^13^C-NMR (151 MHz, DMSO-*d*_6_) δ 204.93, 124.34 (q, ^1^*J*_FC_ = 278.53 Hz, CF_3_), 57.48, 44.56, 43.65 (q, ^2^*J*_FC_ = 34.09 Hz), 25.18. EIMS calcd. for C_8_H_15_F_3_NO_3_S ([M + H]^+^): 262.07, found 260.00.

*Data for N-cyclopropyl-3,3-dimethyl-2-oxobutane-1-sulfonamide* (**P-2**): Yield 15%; white solid; m.p. 72.0–73.6 °C. ^1^H-NMR (600 MHz, DMSO-*d*_6_) δ 7.47 (d, *J* = 2.8 Hz, 1H, SO_2_-*N*H), 4.48 (s, 2H, SO_2_-CH_2_), 2.52 (m, 1H, *N*-CH), 1.12 (d, *J* = 0.7 Hz, 9H, CH_3_), 0.64–0.54 (m, 4H, CH_2_).^13^C-NMR (151 MHz, DMSO-*d*_6_) δ204.82, 55.34, 44.63, 25.30, 24.09, 5.37. HRMS calcd. for C_9_H_18_NO_3_S ([M + H]^+^): 220.1002, found 220.1000.

*Data for N-butyl-3,3-dimethyl-2-oxobutane-1-sulfonamide* (**P-3**): Yield 47%; white solid; m.p. 59.1–60.7 °C. ^1^H-NMR (600 MHz, DMSO-*d*_6_) δ7.02 (t, *J* = 5.9 Hz, 1H, SO_2_-*N*H), 4.41 (s, 2H, SO_2_-CH_2_), 2.94 (td, *J* = 7.2, 5.9 Hz, 2H, *N*-CH_2_), 1.43 (m, 2H, CH_2_), 1.30 (t, *J* = 7.3 Hz, 2H, CH_2_), 1.10 (s, 9H, CH_3_), 0.87 (t, *J* = 7.4 Hz, 3H, CH_3_).^13^C-NMR (151 MHz, DMSO-*d*_6_) δ 204.99, 55.87, 44.58, 31.47, 28.22, 25.28, 19.26, 13.52. HRMS calcd. for C_10_H_22_NO_3_S ([M + H]^+^): 236.1315, found 236.1319.

*Data for 3,3-dimethyl-2-oxo-N-pentylbutane-1-sulfonamide* (**P-4**): Yield 51%; white solid; m.p. 52.2–54.2 °C. ^1^H-NMR (600 MHz, DMSO-*d*_6_) δ 7.02 (t, *J* = 5.9 Hz, 1H, SO_2_-*N*H), 4.41 (s, 2H, SO_2_-CH_2_), 2.93 (td, *J* = 7.2, 6.0 Hz, 2H, *N*-CH_2_), 1.45 (q, *J* = 7.2 Hz, 2H, CH_2_), 1.27 (m, 4H, CH_2_), 1.10 (s, 9H, CH_3_), 0.86 (t, *J* = 6.8 Hz, 3H, CH_3_). ^13^C-NMR (151 MHz, DMSO-*d*_6_) δ 206.23, 54.27, 45.34, 43.72, 29.48, 28.68, 25.63, 22.19, 13.91. HRMS calcd. for C_11_H_24_NO_3_S ([M + H]^+^): 250.1471, found 250.1479.

*Data for N-hexyl-3,3-dimethyl-2-oxobutane-1-sulfonamide* (**P-5**): Yield 22%; white solid; m.p. 59.2–60.2 °C. ^1^H-NMR (600 MHz, DMSO-*d*_6_) δ 7.02 (t, *J* = 5.9 Hz, 1H, SO_2_-*N*H), 4.41 (s, 2H, SO_2_-CH_2_), 2.93 (td, *J* = 7.2, 5.9 Hz, 2H, *N*-CH_2_), 1.43 (q, *J* = 7.2 Hz, 2H, CH_2_), 1.35–1.17 (m, 6H, CH_2_), 1.10 (s, 9H, CH_3_), 0.86 (t, *J* = 6.9 Hz, 3H, CH_3_).^13^C-NMR (151 MHz, DMSO-*d*_6_) δ 204.97, 55.90, 44.57, 42.59, 30.83, 29.35, 25.76, 25.28, 22.01, 13.87. HRMS calcd. for C_12_H_26_NO_3_S ([M + H]^+^): 264.1628, found 264.1629.

*Data for N-(2-fluorophenyl)-3,3-dimethyl-2-oxobutane-1-sulfonamide* (**P-6**): Yield 79%; white solid; m.p. 126.8–128.6 °C. ^1^H-NMR (600 MHz, DMSO-*d*_6_) δ 9.75 (s, 1H, SO_2_-*N*H), 7.61 (dd, *J* = 9.9, 2.1 Hz, 2H, C_6_H_4_-H), 7.43–7.35 (m, 2H, C_6_H_4_-H), 4.61 (s, 2H, SO_2_-CH_2_), 1.08 (s, 9H, CH^3^). ^13^C *N*MR (151 MHz, DMSO-*d*_6_) δ 204.24, 155.39 (d, ^1^*J_FC_* = 251.79 Hz, F-Ph), 127.77 (d, ^3^*J_FC_* = 3.64 Hz), 127.68, 124.53 (d, ^2^*J_FC_* = 12.64 Hz), 119.38 (d, ^2^*J_FC_* = 23.46 Hz), 117.84 (d, ^3^*J_FC_* = 8.89 Hz), 57.05, 44.62, 25.23. HRMS calcd. for C_12_H_17_FNO_3_S ([M + H]^+^): 274.0908, found 274.0907.

*Data for N-(3-fluorophenyl)-3,3-dimethyl-2-oxobutane-1-sulfonamide (***P-7***):* Yield 50%; white solid; m.p. 100.9–102.0 °C. ^1^H-NMR (600 MHz, DMSO-*d*_6_) δ 10.17 (s, 1H, SO_2_-*N*H), 7.35 (m, 1H, C_6_H_4_-H), 7.08–7.00 (m, 2H, C_6_H_4_-H), 6.94–6.88 (m, 1H, C_6_H_4_-H), 4.59 (s, 2H, SO_2_-CH_2_), 1.04 (s, 9H, CH_3_). ^13^C-NMR (151 MHz, DMSO-*d*_6_) δ 205.89, 163.12 (d, ^1^*J_FC_* = 247.2 Hz, F-Ph), 138.07 (d, ^2^*J_FC_* = 10.2 Hz), 130.80 (d, ^3^*J_FC_* = 9.41 Hz), 117.48 (d, ^3^*J_FC_* = 3.03 Hz), 112.98 (d, ^2^*J_FC_* = 21.32 Hz), 109.43 (d, ^2^*J_FC_* = 25.35 Hz), 53.15, 45.44, 25.52. HRMS calcd. for C_12_H_17_FNO_3_S ([M + H]^+^): 274.0908, found 274.0907.

*Data for N-(4-fluorophenyl)-3,3-dimethyl-2-oxobutane-1-sulfonamide* (**P-8**): Yield 51%; white solid; m.p. 125.1–125.6 °C. ^1^H-NMR (600 MHz, DMSO-*d*_6_) δ 9.85 (s, 1H, SO_2_-*N*H), 7.30- 7.23 (m, 2H, C_6_H_4_-H_4_), 7.19 (t, *J* = 8.8 Hz, 2H, C_6_H_4_-H), 4.46 (s, 2H, SO_2_-CH_2_), 1.05 (s, 9H, CH_3_). ^13^C-NMR (151 MHz, DMSO-*d*_6_) δ 206.19, 161.10 (d, ^1^*J_FC_* = 246.23 Hz, F-Ph), 132.30 (d, ^3^*J_FC_* = 3.02 Hz), 124.97 (d, ^3^*J_FC_* = 8.36 Hz), 116.38 (d, ^2^*J_FC_* = 22.71 Hz), 52.63, 45.45, 25.54. HRMS calcd. for C_12_H_17_FNO_3_S ([M + H]^+^): 274.0908, found 274.0905.

*Data for N-(2-chlorophenyl)-3,3-dimethyl-2-oxobutane-1-sulfonamide* (**P-9**): Yield 15%; white solid; m.p. 80.4–82.3 °C. ^1^H-NMR (600 MHz, DMSO-*d*_6_) δ 9.37 (s, 1H, SO_2_-*N*H), 7.51 (m, 2H, C_6_H_4_-H), 7.35 (td, *J* = 7.7, 1.5 Hz, 1H, C_6_H_4_-H), 7.26 (td, *J* = 7.7, 1.6 Hz, 1H, C_6_H_4_-H), 4.63 (s, 2H, SO_2_-CH_2_), 1.08 (s, 9H, CH_3_). ^13^C-NMR (151 MHz, DMSO-*d*_6_) δ 204.29, 133.81, 129.84, 128.71, 127.81, 127.39, 127.22, 57.78, 44.59, 25.31. HRMS calcd. for C_12_H_17_ClNO_3_S ([M + H]^+^): 290.0612, found 290.0614.

*Data for N-(3-chlorophenyl)-3,3-dimethyl-2-oxobutane-1-sulfonamide* (**P-10**): Yield 40%; white solid; m.p. 108.7–111.5 °C. ^1^H-NMR (600 MHz, DMSO-*d*_6_) δ 10.16 (s, 1H, SO_2_-*N*H), 7.36 (t, *J* = 8.1 Hz, 1H, C_6_H_4_-H), 7.27 (d, *J* = 2.1 Hz, 1H, C_6_H_4_-H), 7.20 (m, 1H, C_6_H_4_-H), 7.16 (m, 1H, C_6_H_4_-H), 4.59 (s, 2H, SO_2_-CH_2_), 1.05 (s, 9H, CH_3_). ^13^C-NMR (151 MHz, DMSO-*d*_6_) δ 205.94, 137.67, 135.20, 130.59, 126.34, 122.23, 120.32, 53.14, 45.46, 25.53. HRMS calcd. for C_12_H_17_ClNO_3_S ([M + H]^+^): 290.0612, found 290.0603.

*Data for N-(4-chlorophenyl)-3,3-dimethyl-2-oxobutane-1-sulfonamide* (**P-11**): Yield 45%; white solid; m.p. 112.4–114.1 °C. ^1^H-NMR (600 MHz, DMSO-*d*_6_) δ 10.03 (s, 1H, SO_2_-*N*H), 7.41–7.36 (m, 2H, C_6_H_4_-H), 7.26–7.22 (m, 2H, C_6_H_4_-H), 4.52 (s, 2H, SO_2_-CH_2_), 1.04 (s, 9H, CH_3_). ^13^C-NMR (151 MHz, DMSO-*d*_6_) δ 206.03, 134.99, 131.97, 129.69, 123.89, 77.26, 77.05, 76.84, 52.81, 45.46, 25.53. HRMS calcd. for C_12_H_17_ClNO_3_S ([M + H]^+^): 290.0612, found 290.0609.

*Data for N-(2-bromophenyl)-3,3-dimethyl-2-oxobutane-1-sulfonamide* (**P-12**): Yield 31%; white solid; m.p. 92.9–94.1 °C. ^1^H-NMR (600 MHz, DMSO-*d*_6_) δ 9.26 (s, 1H, SO_2_-*N*H), 7.68 (dd, *J* = 8.0, 1.4 Hz, 1H, C_6_H_4_-H), 7.50 (dd, *J* = 8.0, 1.6 Hz, 1H, C_6_H_4_-H), 7.40 (td, *J* = 7.7, 1.5 Hz, 1H, C_6_H_4_-H), 7.19 (td, *J* = 7.7, 1.6 Hz, 1H, C_6_H_4_-H), 4.64 (s, 2H, SO_2_-CH_2_), 1.09 (s, 9H, CH_3_). ^13^C-NMR (151 MHz, DMSO-*d*_6_) δ 204.32, 135.16, 133.09, 128.43, 127.83, 127.46, 119.76, 57.98, 44.61, 25.32. EIMS calcd. for C_12_H_17_BrNO_3_S ([M + H]^+^): 334.01, found 333.90.

*Data for N-(3-bromophenyl)-3,3-dimethyl-2-oxobutane-1-sulfonamide* (**P-13**): Yield 26%; white solid; m.p. 114.5–115.5 °C. ^1^H-NMR (600 MHz, DMSO-*d*_6_) δ 10.13 (s, 1H, SO_2_-*N*H), 7.40 (d, *J* = 2.2 Hz, 1H, C_6_H_4_-H), 7.32–7.25 (m, 1H, C_6_H_4_-H), 7.28 (s, 1H, C_6_H_4_-H), 7.24 (dt, *J* = 6.0, 2.5 Hz, 1H, C_6_H_4_-H), 4.58 (s, 2H, SO_2_-CH_2_), 1.04 (s, 9H, CH_3_). ^13^C-NMR (151 MHz, DMSO-*d*_6_) δ 204.20, 139.59, 131.11, 126.36, 121.83, 121.79, 118.26, 55.36, 44.72, 25.06. HRMS calcd. for C_12_H_17_BrNO_3_S ([M + H]^+^): 334.0107, found 334.0108.

*Data for N-(4-bromophenyl)-3,3-dimethyl-2-oxobutane-1-sulfonamide* (**P-14**): Yield 38%; white solid; m.p. 112.1–113.5 °C. ^1^H-NMR (600 MHz, DMSO-*d*_6_) δ 10.04 (s, 1H, SO_2_-*N*H), 7.54–7.48 (m, 2H, C_6_H_4_-H), 7.21–7.15 (m, 2H, C_6_H_4_-H), 4.53 (s, 2H, SO_2_-CH_2_), 1.04 (s, 9H, CH_3_). ^13^C-NMR (151 MHz, DMSO-*d*_6_) δ 204.19, 137.34, 131.95, 121.65, 115.88, 55.10, 44.70, 25.08. EIMS calcd. for C_12_H_17_BrNO_3_S ([M + H]^+^): 334.01, found 333.90.

*Data for N-(2-methoxyphenyl)-3,3-dimethyl-2-oxobutane-1-sulfonamide* (**P-15**): Yield 14%; white solid; m.p. 70.0–71.1 °C. ^1^H-NMR (600 MHz, DMSO-*d*_6_) δ 8.72 (s, 1H, SO_2_-*N*H), 7.31 (dd, *J* = 7.9, 1.6 Hz, 1H, C_6_H_4_-H), 7.22–7.16 (m, 1H, C_6_H_4_-H), 7.07 (dd, *J* = 8.2, 1.3 Hz, 1H, C_6_H_4_-H), 6.94 (td, *J* = 7.6, 1.3 Hz, 1H, C_6_H_4_-H), 4.54 (s, 2H, SO_2_-CH_2_), 3.82 (s, 3H), 1.07 (s, 9H, CH_3_). ^13^C-NMR (151 MHz, DMSO-*d*_6_) δ 204.39, 149.67, 126.04, 125.65, 121.29, 120.82, 111.10, 56.00, 54.17, 45.13, 25.71. HRMS calcd. for C_13_H_20_NO_4_S ([M + H]^+^): 286.1108, found 286.1109.

*Data for 3,3-dimethyl-2-oxo-N-(2-(trifluoromethyl)phenyl)butane-1-sulfonamide* (**P-16**): Yield 61%; white solid; m.p. 78.3–79.3 °C. ^1^H-NMR (600 MHz, DMSO-*d*_6_) δ 9.38 (s, 1H, SO_2_-*N*H), 7.75 (dd, *J* = 7.9, 1.5 Hz, 1H, C_6_H_4_-H), 7.71 (td, *J* = 7.7, 1.5 Hz, 1H, C_6_H_4_-H), 7.65 (d, *J* = 8.0 Hz, 1H, C_6_H_4_-H), 7.50 (t, *J* = 7.7 Hz, 1H, C_6_H_4_-H), 4.70 (s, 2H, SO_2_-CH_2_), 1.13 (s, 9H, CH_3_). ^13^C-NMR (151 MHz, DMSO-*d*_6_) δ 204.67, 133.97, 133.19, 126.65 (d, ^3^*J_FC_* = 5.03 Hz), 125.44, 123.69 (d, ^1^*J_FC_* = 273.11 Hz, CF_3_), 121.83 (d, ^2^*J_FC_* = 29.82 Hz), 56.57, 45.19, 25.74. HRMS calcd. for C_13_H_17_F_3_NO_3_S ([M + H]^+^): 324.0876, found 324.0866.

*Data for 3,3-dimethyl-2-oxo-N-(3-(trifluoromethyl)phenyl)butane-1-sulfonamide* (**P-17**): Yield 65%; white solid; m.p. 134.0–136.0 °C. ^1^H-NMR (600 MHz, DMSO-*d*_6_) δ10.31 (s, 1H, SO_2_-*N*H), 7.58 (t, *J* = 8.2 Hz, 1H, C_6_H_4_-H), 7.56–7.51 (m, 2H, C_6_H_4_-H), 7.48–7.43 (m, 1H, C_6_H_4_-H), 4.62 (s, 2H, SO_2_-CH_2_), 1.04 (s, 9H, CH_3_). ^13^C-NMR (151 MHz, DMSO-*d*_6_) δ 206.01, 137.16, 132.06 (d, ^2^*J_FC_* = 32.9 Hz), 130.23, 125.43, 123.52 (d, ^1^*J_FC_* = 272,87 Hz, CF_3_)122.78 (d, ^3^*J_FC_* = 3.9 Hz), 118.83 (d, ^3^*J_FC_* = 3.8 Hz), 53.42, 45.47, 25.50. HRMS calcd. for C_13_H_17_F_3_NO_3_S ([M + H]^+^): 324.0876, found 324.0877.

*Data for 3,3-dimethyl-2-oxo-N-(4-(trifluoromethoxy)phenyl)butane-1-sulfonamide* (**P-18**): Yield 27%; white solid; m.p. 119.3–120.1 °C. ^1^H-NMR (600 MHz, DMSO-*d*_6_) δ 10.12 (s, 1H, SO_2_-*N*H), 7.38–7.30 (m, 4H, C_6_H_4_-H), 4.56 (s, 2H, SO_2_-CH_2_), 1.04 (s, 9H, CH_3_).^13^C-NMR (151 MHz, DMSO-*d*_6_) δ 204.27, 144.31, 137.13, 122.08, 121.12, 120.07 (q, ^1^*J_FC_* = 256.08 Hz, CF_3_), 55.26, 44.69, 25.04. HRMS calcd. for C_13_H_17_F_3_NO_4_S ([M + H]^+^): 340.0825, found 340.0833.

*Data for N-(4-chloro-2-fluorophenyl)-3,3-dimethyl-2-oxobutane-1-sulfonamide* (**P-19**): Yield 71%; white solid; m.p. 115.3–117.1 °C. ^1^H-NMR (600 MHz, DMSO-*d*_6_) δ 9.75 (s, 1H, SO_2_-*N*H), 7.51 (dd, *J* = 10.3, 2.4 Hz, 1H, C_6_H_4_-H), 7.44 (t, *J* = 8.6 Hz, 1H, C_6_H_4_-H), 7.28 (m, 1H, C_6_H_4_-H), 4.61 (s, 2H, SO_2_-CH_2_), 1.08 (s, 9H, CH_3_). ^13^C-NMR (151 MHz, DMSO-*d*_6_) δ 204.23, 155.45 (d, ^1^*J_FC_* = 251.09 Hz, F-Ph), 130.24 (d, ^3^*J_FC_* = 9.60 Hz), 127.51, 124.85 (d, ^3^*J_FC_* = 3.59 Hz), 124.08 (d, ^2^*J_FC_* = 12.73 Hz), 116.65 (d, ^2^*J_FC_* = 23.80 Hz), 57.06, 44.61, 25.23. HRMS calcd. for C_12_H_16_ClFNO_3_S ([M + H]^+^): 308.0518, found 308.0516.

*Data for N-(4-bromo-2-fluorophenyl)-3,3-dimethyl-2-oxobutane-1-sulfonamide* (**P-20**): Yield 55%; white solid; m.p. 113.5–114.7 °C. ^1^H-NMR (600 MHz, DMSO-*d*_6_) δ 9.76 (s, 1H, SO_2_-*N*H), 7.62 (dd, *J* = 10.0, 2.0 Hz, 1H, C_6_H_4_-H), 7.45–7.37 (m, 2H, C_6_H_4_-H), 4.60 (d, *J* = 27.3 Hz, 2H, SO_2_-CH_2_), 1.08 (d, *J* = 0.9 Hz, 9H, CH_3_).^13^C-NMR (151 MHz, DMSO-*d*_6_) δ 205.04, 154.12 (d, ^1^*J_FC_* = 250.99 Hz, F-Ph), 128.25 (d, ^3^*J_FC_* = 3.70 Hz), 125.30, 123.85 (d, ^2^*J_FC_* = 12.50 Hz), 119.52 (d, ^2^*J_FC_* = 22.65 Hz), 118.83 (d, ^3^*J_FC_* = 8.89 Hz), 54.60, 45.33, 25.63. HRMS calcd. for C_12_H_16_BrFNO_3_S ([M + H]^+^):352.0013, found 352.0012.

*Data for N-(2-fluoro-5-(trifluoromethyl)phenyl)-3,3-dimethyl-2-oxobutane-1-sulfonamide* (**P-21**): Yield 63%; white solid; m.p. 113.7–115.1 °C. ^1^H-NMR (600 MHz, DMSO-*d*_6_) δ 10.04 (s, 1H, SO_2_-*N*H), 7.76 (dd, *J* = 7.2, 2.3 Hz, 1H, C_6_H_4_-H), 7.64 (dt, *J* = 7.0, 3.2 Hz, 1H, C_6_H_4_-H), 7.53 (t, *J* = 9.3 Hz, 1H, C_6_H_4_-H), 4.72 (s, 2H, SO_2_-CH_2_), 1.09 (s, 9H, CH_3_). ^13^C-NMR (151 MHz, DMSO-*d*_6_) δ 204.25, 157.02 (d, ^1^*J_FC_* = 253.Hz, F-Ph), 126.09 (d, ^2^*J_FC_* = 13.74 Hz), 125.46 (dq, ^2^*J_FC_* = 32.46 Hz), 123.87 (q, ^3^*J_FC_* = 4.06 Hz), 123.53 (q, ^1^*J_FC_* = 272.02 Hz, CF_3_), 122.40 (q, ^3^*J_FC_* = 3.40 Hz) 117.32 (d, ^2^*J_FC_* = 21.62 Hz), 57.46, 44.62, 25.16. HRMS calcd. for C_13_H_16_F_4_NO_3_S ([M + H]^+^): 342.0782, found 342.0780.

*Data for N-(4-bromo-3-fluorophenyl)-3,3-dimethyl-2-oxobutane-1-sulfonamide* (**P-22**): Yield 52%; white solid; m.p. 116.0–117.1 °C. ^1^H-NMR (600 MHz, DMSO-*d*_6_) δ 10.32 (s, 1H, SO_2_-*N*H), 7.65 (t, *J* = 8.4 Hz, 1H, C_6_H_4_-H), 7.19 (dd, *J* = 10., 8 2.5 Hz, 1H, C_6_H_4_-H), 7.02 (dd, *J* = 8.7, 2.5 Hz, 1H, C_6_H_4_-H), 4.66 (s, 2H, SO_2_-CH_2_), 1.05 (s, 9H, CH_3_). ^13^C-NMR (151 MHz, DMSO-*d*_6_) δ 204.61, 158.66 (d, ^1^*J_FC_* = 243.74 Hz, F-Ph), 139.75 (d, ^3^*J_FC_* = 9.93 Hz), 134.12, 117.09 (d, ^3^*J_FC_* = 3.22 Hz), 107.71 (d, ^2^*J_FC_* = 26.32 Hz), 101.95 (d, ^2^*J_FC_* = 20.88 Hz), 55.87, 45.12, 25.41. EIMS calcd. for C_12_H_16_BrFNO_3_S ([M + H]^+^): 352.00, found 351.90.

*Data for N-(4-chloro-2-(trifluoromethyl)phenyl)-3,3-dimethyl-2-oxobutane-1-sulfonamide* (**P-23**): Yield 63%; white solid; m.p. 127.4–128.7 °C. ^1^H-NMR (600 MHz, DMSO-*d*_6_) δ 9.50 (s, 1H, SO_2_-*N*H), 7.80 (s, 1H, C_6_H_4_-H), 7.83–7.76 (m, 1H, C_6_H_4_-H), 7.66 (dd, *J* = 8.7, 3.2 Hz, 1H, C_6_H_4_-H), 4.71 (d, *J* = 3.0 Hz, 2H, SO_2_-CH_2_), 1.10 (d, *J* = 3.8 Hz, 9H, CH_3_). ^13^C-NMR (151 MHz, DMSO-*d*_6_) δ 204.43, 133.25, 133.19, 131.46, 131.29, 127.22 (q, ^2^*J_FC_* = 30.17 Hz), 126.70 (q, ^3^*J_FC_* = 5.27 Hz), 122.46 (q, ^1^*J_FC_* = 274.11 Hz, CF_3_), 58.52, 44.60, 25.27. HRMS calcd. for C_13_H_16_ClF_3_NO_3_S ([M + H]^+^): 358.0486, found 358.0487.

*Data for N-(2-chloro-5-(trifluoromethyl)phenyl)-3,3-dimethyl-2-oxobutane-1-sulfonamide* (**P-24**): Yield 9%; white solid; m.p. 97.3–98.5 °C. ^1^H-NMR (600 MHz, DMSO-*d*_6_) δ 9.75 (s, 1H, SO_2_-*N*H), 7.82 – 7.75 (m, 2H, C_6_H_4_-H), 7.62 (dd, *J* = 8.4, 2.1 Hz, 1H, C_6_H_4_-H), 4.77 (s, 2H, SO_2_-CH_2_), 1.09 (s, 9H, CH_3_). ^13^C-NMR (151 MHz, DMSO-*d*_6_) δ 204.40, 135.02, 132.39, 131.04, 128.30 (q, ^2^*J_FC_* = 32.39 Hz), 123.49 (q, ^3^*J_FC_* = 3.83 Hz), 123.44 (q, ^1^*J_FC_* = 272.52 Hz, CF_3_), 122.89 (q, ^3^*J_FC_* = 3.96 Hz), 58.36, 44.57, 25.25. HRMS calcd. for C_13_H_16_ClF_3_NO_3_S ([M + H]^+^): 358.0486, found 358.0487.

*Data for N-(4-bromo-2-nitrophenyl)-3,3-dimethyl-2-oxobutane-1-sulfonamide* (**P-25**): Yield 35%; white solid; m.p. 102.4–106.0 °C. ^1^H-NMR (600 MHz, DMSO-*d*_6_) δ 9.89 (s, 1H, SO_2_-*N*H), 8.25 (d, *J* = 2.3 Hz, 1H, C_6_H_4_-H), 7.96 (dd, *J* = 8, 82.3 Hz, 1H, C_6_H_4_-H), 7.63 (d, *J* = 8.8 Hz, 1H, C_6_H_4_-H), 4.83 (s, 2H, SO_2_-CH_2_), 1.07 (s, 9H, CH_3_). ^13^C-NMR (151 MHz, DMSO-*d*_6_) δ 204.75, 142.18, 137.27, 130.61, 127.88, 126.86, 116.95, 57.53, 44.63, 25.12. HRMS calcd. for C_12_H_16_BrN_2_O_5_S([M + H]^+^ ): 378.9958, found 378.9957.

*Data for N-(3-bromo-4-fluorophenyl)-3,3-dimethyl-2-oxobutane-1-sulfonamide* (**P-26**): Yield 68%; white solid; m.p. 146.9–148.7 °C. ^1^H-NMR (600 MHz, DMSO-*d*_6_) δ 10.06 (s, 1H, SO_2_-*N*H), 7.50 (dd, *J* = 6.1, 2.6 Hz, 1H, C_6_H_4_-H), 7.37 (t, *J* = 8.8 Hz, 1H, C_6_H_4_-H), 7.27 (ddd, *J* = 8.9, 4.3, 2.7 Hz, 1H, C_6_H_4_-H), 4.57 (s, 2H, SO_2_-CH_2_), 1.06 (s, 9H, CH_3_). ^13^C-NMR (151 MHz, DMSO-*d*_6_) δ 204.37, 155.13 (d, ^1^*J_FC_* = 241.60 Hz, F-Ph), 135.25 (d, ^3^*J_FC_* = 2.97 Hz), 124.57, 121.39 (d, ^3^*J_FC_* = 7.49 Hz), 117.10 (d, ^2^*J_FC_* = 23.24 Hz), 108.04 (d, ^2^*J_FC_* = 22.11 Hz), 55.27, 44.71, 25.06. EIMS calcd. for C_12_H_16_BrFNO_3_S ([M + H]^+^): 352.00, found 351.90.

*Data for N-(4-bromo-3-methylphenyl)-3,3-dimethyl-2-oxobutane-1-sulfonamide* (**P-27**): Yield 30%; white solid; m.p. 133.9–134.5 °C. ^1^H-NMR (600 MHz, DMSO-*d*_6_) δ 9.97 (s, 1H, SO_2_-*N*H), 7.51 (d, *J* = 8.6 Hz, 1H, C_6_H_4_-H), 7.18 (d, *J* = 2.6 Hz, 1H, C_6_H_4_-H), 7.00 (dd, *J* = 8.6, 2.7 Hz, 1H, C_6_H_4_-H), 4.52 (s, 2H, SO_2_-CH_2_), 2.30 (s, 3H, ph-CH_3_), 1.04 (s, 9H, CH_3_). ^13^C-NMR (151 MHz, DMSO-*d*_6_) δ 206.11, 139.85, 139.36, 134.53, 123.88, 121.06, 120.38, 57.00, 46.62, 27.00, 24.54. EIMS calcd. for C_13_H_19_BrNO_3_S ([M + H]^+^): 348.02, found 347.90.

*Data for 3,3-dimethyl-2-oxo-N-(2,4,5-trifluorophenyl)butane-1-sulfonamide* (**P-28**): Yield 21%; white solid; m.p. 114.1–115.4°C. ^1^H-NMR (600 MHz, DMSO-*d*_6_) δ 9.84 (s, 1H, SO_2_-*N*H), 7.65 (td, *J* = 10.4, 7.3 Hz, 1H, C_6_H_4_-H), 7.51 (dt, *J* = 11.5, 7.9 Hz, 1H, C_6_H_4_-H), 4.66 (s, 2H, SO_2_-CH_2_), 1.08 (s, 9H, CH_3_). ^13^C-NMR (151 MHz, DMSO-*d*_6_) δ 205.13, 148.28 (qd, ^1^*J_FC_* = 244.26 Hz, F-Ph), 147.69 (dq, ^1^*J_FC_* = 250,84 Hz, F-Ph) 120.63–120.51 (m), 113.34 (d, ^2^*J_FC_* = 19.70 Hz), 105.79 (d, ^2^*J_FC_* = 21.79 Hz), 105.70 (d, ^2^*J_FC_* = 21.85), 54.81, 45.35, 25.60. HRMS calcd. for C_12_H_15_Cl_3_NO_3_S ([M + H]^+^): 310.0719, found 310.0719.

*Data for 3,3-dimethyl-2-oxo-N-(2,4,5-trichlorophenyl)butane-1-sulfonamide* (**P-29**): Yield 62%; white solid; m.p. 144.3–115.2 °C. ^1^H-NMR (600 MHz, DMSO-*d*_6_) δ 9.70 (s, 1H, SO_2_-*N*H), 7.93 (s, 1H, C_6_H_4_-H), 7.73 (s, 1H, C_6_H_4_-H), 4.78 (s, 2H, SO_2_-CH_2_), 1.10 (s, 9H, CH_3_). ^13^C-NMR (151 MHz, DMSO-*d*_6_) δ 207.17, 137.04, 133.50, 132.77, 131.39, 130.59, 130.02, 61.02, 47.31, 28.00. HRMS calcd. for C_12_H_15_F_3_NO_3_S ([M + H]^+^): 357.9833, found 357.9829.

*Data for 3,3-dimethyl-2-oxo-N-(2,4,6-tribromophenyl)butane-1-sulfonamide* (**P-30**): Yield 9%; white solid; m.p. 135.6–137.6 °C. ^1^H-NMR (600 MHz, DMSO-*d*_6_) δ 9.61 (s, 1H, SO_2_-*N*H), 8.02 (s, 2H, C_6_H_4_-H), 4.78 (s, 2H, SO_2_-CH_2_), 1.13 (s, 9H, CH_3_). ^13^C-NMR (151 MHz, DMSO-*d*_6_) δ 204.94, 202.64, 136.08, 135.09, 134.83, 129.43, 127.72, 122.06, 61.57, 44.89, 25.94, 25.49. HRMS calcd. for C_12_H_15_Br_3_NO_3_S ([M + H]^+^): 489.8317, found 489.8315.

*Data for 3,3-dimethyl-2-oxo-N-phenethylbutane-1-sulfonamide* (**P-31**): Yield 22%; white solid; m.p. 73.6–74.4 °C. ^1^H-NMR (600 MHz, DMSO-*d*_6_) δ 7.30 (t, *J* = 7.6 Hz, 2H, C_6_H5-H), 7.26 – 7.19 (m, 3H, C_6_H5-H), 7.17 (t, *J* = 5.8 Hz, 1H, SO_2_-*N*H)), 4.40 (s, 2H, SO_2_-CH_2_), 3.23–3.18 (m, 2H, *N*-CH_2_), 2.77 (dd, *J* = 8.7, 6.7 Hz, 2H, CH_2_), 1.08 (s, 9H, CH_3_). ^13^C-NMR (151 MHz, DMSO-*d*_6_) δ 204.99, 138.89, 128.70, 128.30, 126.21, 56.07, 44.58, 44.13, 35.71, 25.24. HRMS calcd. for C_14_H_22_NO_3_S ([M + H]^+^): 284.1315, found 284.1319.

*Data for 3,3-dimethyl-N-(4-nitrophenethyl)-2-oxobutane-1-sulfonamide* (**P-32**): Yield 9%; yellow solid; m.p. 92.3–98.5 °C. ^1^H-NMR (600 MHz, DMSO-*d*_6_) δ 8.18 (d, 2H, C_6_H_4_-2H), 7.55 (d,1H, 2H, C_6_H_4_-2H), 7.24 (s, 1H, SO_2_-*N*H), 4.45 (s, 2H, CH_2_-SO_2_), 3.33 (s, 2H, *N*H-CH_2_), 2.93 (t, 2H, C_6_H_4_-CH_2_), 1.09 (s, 9H, 3CH_3_). ^13^C-NMR (151 MHz, CDCl_3_) δ 205.98, 146.90, 145.51, 129.70, 129.01, 128.10, 123.82, 54.74, 45.30, 44.00, 36.28, 26.84, 25.48. HRMS calcd. for C_14_H_21_N_2_O_5_S ([M + H]^+^): 329.1171, found 329.1166.

*Data for 3,3-dimethyl-N-(4-methylphenethyl)-2-oxobutane-1-sulfonamide* (**P-33**): Yield 24%; white solid; m.p. 78.6–79.8 °C. ^1^H-NMR (600 MHz, DMSO-*d*_6_) δ 7.13 (m, 5H, C_6_H_4_ + SO_2_-*N*H), 4.37 (s, 2H, CH_2_-SO_2_), 3.18 (m, 2H, *N*H-CH_2_), 2.74 (t, 2H, C_6_H_4_-CH_2_), 2.27 (s, 3H, C_6_H_4_-CH_3_), 1.09 (d, 9H, 3CH_3_).^13^C-NMR (151 MHz, CDCl_3_) δ 205.75, 136.29, 134.68, 129.33, 128.64, 54.78, 45.18, 44.87, 35.84, 25.52, 20.95. HRMS calcd. for C_15_H_23_NO_3_NaS ([M + Na]^+^): 320.1296, found 320.1297.

*Data for N-(4-methoxyphenethyl)-3,3-dimethyl-2-oxobutane-1-sulfonamide* (**P-34**): Yield 11%; white solid; m.p. 88.9–89.8 °C. ^1^H-NMR (600 MHz, DMSO-*d*_6_) δ 7.14 (t, 3H, C_6_H_4_-3H), 6.86 (d, 2H, C_6_H_4_-H + SO_2_-*N*H), 4.36 (s, 2H, CH_2_-SO_2_), 3.72 (s, 3H, CH_3_O), 3.15 (m, 2H, *N*H-CH_2_), 2.70 (t, 2H, C_6_H_4_-CH_2_), 1.08 (s, 9H, 3CH_3_). ^13^C-NMR (151 MHz, CDCl_3_) δ 205.85, 158.51, 129.83, 129.79, 114.17, 55.28, 54.77, 45.29, 45.06, 35.48, 25.62. HRMS calcd. for C_15_H_23_NO_4_NaS ([M + Na]^+^): 336.1245, found 336.1242.

*Data for N-(3-methoxyphenethyl)-3,3-dimethyl-2-oxobutane-1-sulfonamide* (**P-35**): Yield 5%; white solid; m.p. 52.3–53.8 °C. ^1^H-NMR (600 MHz, DMSO-*d*_6_) δ 7.21 (t, 1H, C_6_H_4_-H), 7.15 (t, 1H, C_6_H_4_-H), 6.81–6.77 (m, 3H, C_6_H_4_-2H + SO_2_-*N*H), 4.39 (s, 2H, CH_2_-SO_2_), 3.74 (s, 3H, CH_3_O), 3.20 (m, 2H, *N*H-CH_2_), 2.74 (t, 2H, C_6_H_4_-CH_2_), 1.08 (s, 9H, 3CH_3_). ^13^C-NMR (151 MHz, CDCl_3_) δ 205.76, 159.76, 139.34, 129.67, 120.98, 114.39, 112.23, 55.10, 54.78, 45.19, 44.65, 36.33, 25.51. HRMS calcd. for C_15_H_23_NO_4_NaS ([M + Na]^+^): 336.1245, found 336.1248.

*Data for N-(3,4-dimethoxyphenethyl)-3,3-dimethyl-2-oxobutane-1-sulfonamide* (**P-36**): Yield 3%; yellow solid; m.p. 93.8–95.6 °C. ^1^H-NMR (600 MHz, DMSO-*d*_6_) δ 7.12 (t, 1H, C_6_H_4_-H), 6.85 (m, 2H, C_6_H_4_-2H), 6.74 (dd, 1H, C_6_H_4_-H), 4.36 (s, 2H, CH_2_-SO_2_), 3.73 (d, 6H, 2CH_3_O), 3.19 (m, 2H, *N*H-CH_2_), 2.70 (t, 2H, C_6_H_4_-CH_2_), 1.07 (s, 9H, 3CH_3_). ^13^C-NMR (151 MHz, CDCl_3_) δ 205.79, 148.94, 147.77, 130.30, 120.69, 111.95, 111.34, 55.82, 55.77, 54.85, 45.17, 44.83, 35.83, 25.47. HRMS calcd. for C_16_H_25_NO_5_NaS ([M + Na]^+^): 366.1351, found 366.1054.

*Data for N-(4-fluorophenethyl)-3,3-dimethyl-2-oxobutane-1-sulfonamide* (**P-37**): Yield 10%; yellow solid; m.p. 81.0–83.6 °C. ^1^H-NMR (600 MHz, DMSO-*d*_6_) δ 7.28 (dd, 2H, C_6_H_4_-2H), 7.17 (t, 1H, SO_2_-*N*H), 7.13 (t, 2H, C_6_H_4_-2H), 4.41 (s, 2H, CH_2_-SO_2_), 3.20 (m, 2H, *N*H-CH_2_), 2.77 (t, 2H, C_6_H_4_-CH_2_), 1.09 (d, 9H, 3CH_3_). ^13^C-NMR (151 MHz, CDCl_3_) δ 205.86, 162.51, 160.89 (s,^1^*J* = 244.62), 133.44, 130.22, 130.21, 115.52, 115.38, 54.71, 45.23, 44.76, 35.52, 25.50. HRMS calcd. for C_14_H_20_FNO_3_NaS ([M + Na]^+^): 324.1046, found 324.1040.

*Data for N-(3-fluorophenethyl)-3,3-dimethyl-2-oxobutane-1-sulfonamide* (**P-38**): Yield 7%; yellow solid; m.p. 47.4–52.2 °C. ^1^H-NMR (600 MHz, DMSO-*d*_6_) δ 7.34 (td, 1H, C_6_H_4_-H), 7.18 (t, 1H, C_6_H_4_-H), 7.09 (m, 2H, C_6_H_4_-2H), 7.04 (td, 1H, SO_2_-*N*H), 4.42 (s, 2H, CH_2_-SO_2_), 3.23 (td, 2H, *N*H-CH_2_), 2.80 (t, 2H, C_6_H_4_-CH_2_), 1.09 (s, 9H, 3CH_3_). ^13^C-NMR (151 MHz, CDCl_3_) δ 205.85, 163.65, 162.02 (s,^1^*J* = 246.13), 140.37, 140.32, 130.15, 130.10, 124.45, 124.43, 115.73, 115.58, 113.73, 113.59, 54.80, 45.22, 44.43, 36.06, 36.05, 28.36, 25.50. HRMS calcd. for C_14_H_20_FNO_3_NaS ([M + Na]^+^): 324.1046, found 324.1042.

*Data for N-(2-fluorophenethyl)-3,3-dimethyl-2-oxobutane-1-sulfonamide* (**P-39**): Yield 13%; white solid; m.p. 69.8–73.0 °C. ^1^H-NMR (600 MHz, DMSO-*d*_6_) δ 7.34 (td, 1H, C_6_H_4_-H), 7.28 (dq, 2H, C_6_H_4_-2H), 7.16 (m, 2H, C_6_H_4_-H + SO_2_-*N*H), 4.43 (d, 2H, CH_2_-SO_2_), 3.21 (d, 2H, *N*H-CH_2_), 2.82 (t, 2H, C6H4-CH_2_), 1.09 (d, 9H, 3CH_3_). ^13^C-NMR (151 MHz, CDCl_3_) δ 205.79, 161.97, 160.34 (s,^1^*J* = 245.54), 131.18, 131.15, 128.65, 128.60, 124.72, 124.61, 124.23, 124.21, 115.47, 115.32, 54.73, 45.21, 43.39, 43.38, 29.97, 29.96, 25.53. HRMS calcd. for C_14_H_20_FNO_3_NaS ([M + Na]^+^): 324.1046, found 324.1043.

*Data for N-(3-bromophenethyl)-3,3-dimethyl-2-oxobutane-1-sulfonamide* (**P-40**): Yield 22%; yellow solid; m.p. 79.8–81.0 °C. ^1^H-NMR (600 MHz, DMSO-*d*_6_) δ 7.48 (d, 1H, C_6_H_4_-H), 7.42 (m, 1H, C_6_H_4_-H), 7.27 (m, 2H, C_6_H_4_-2H), 7.19 (t, 1H, SO_2_-*N*H), 4.42 (s, 2H, CH_2_-SO_2_), 3.22 (td, 2H, *N*H-CH_2_), 2.78 (t, 2H, C6H4-CH_2_), 1.09 (s, 9H, 3CH_3_). ^13^C-NMR (151 MHz, CDCl_3_) δ 205.94, 140.21, 131.87, 130.30, 130.00, 127.54, 122.72, 54.83, 45.34, 44.55, 36.09, 25.60. HRMS calcd. for C_14_H_20_BrNO_3_NaS ([M + Na]^+^): 384.0245, found 384.0242.

*Data for N-(2-bromobenzyl)-3,3-dimethyl-2-oxobutane-1-sulfonamide* (**P-41**): Yield 33%; yellow solid; m.p. 79.5–83.4 °C. ^1^H-NMR (600 MHz, DMSO-*d*_6_) δ 7.76 (t, 1H, SO_2_-*N*H), 7.61 (dd, 1H, C_6_H_4_-H), 7.55 (m, 1H, C_6_H_4_-H), 7.42 (td, 1H, C_6_H_4_-H), 7.24 (td, 1H, C_6_H_4_-H), 4.55 (s, 2H, CH_2_-SO_2_), 4.28 (d, 2H, *N*H-CH_2_), 1.10 (s, 9H, 3CH_3_). ^13^C-NMR (151 MHz, CDCl_3_) δ 205.73, 135.80, 133.09, 130.65, 129.84, 127.87, 123.87, 55.96, 47.78, 45.09, 25.72. HRMS calcd. for C_13_H_18_BrNO_3_NaS ([M + Na]^+^): 370.0088, found 370.0084.

*Data for N-(2- fluorobenzyl)-3,3-dimethyl-2-oxobutane-1-sulfonamide* (**P-42**): Yield 20%; white solid; m.p. 101.1–103.4 °C. ^1^H-NMR (600 MHz, DMSO-*d*_6_) δ 7.74 (t, 1H, SO_2_-*N*H), 7.54 (m, 1H, C_6_H_4_-H), 7.45 (dd, 1H, C_6_H_4_-H), 7.38 (td, 1H, C_6_H_4_-H), 7.32 (td, 1H, C_6_H_4_-H), 4.54 (s, 2H, CH_2_-SO_2_), 4.30 (d, 2H, *N*H-CH_2_), 1.10 (s, 9H, 3CH_3_). ^13^C-NMR (151 MHz, CDCl_3_) δ 205.68, 134.01, 133.75, 130.42, 129.72, 129.54, 127.15, 55.76, 45.44, 44.99, 25.62. HRMS calcd. for C_13_H_19_FNO_3_S ([M + H]^+^): 288.1070, found 288.1072.

*Data for N-(4-fluorobenzyl)-3,3-dimethyl-2-oxobutane-1-sulfonamide* (**P-43**): Yield 11%; white solid; m.p. 129–133.8 °C. ^1^H-NMR (600 MHz, DMSO-*d*_6_) δ 7.83 (t, 1H, SO_2_-*N*H), 7.73 (d, 2H, C_6_H_4_-2H), 7.58 (d, 2H, C_6_H_4_-2H), 4.49 (s, 2H, CH_2_-SO_2_), 4.30 (d, 2H, *N*H-CH_2_), 1.09 (s, 9H, 3CH_3_). ^13^C-NMR (151 MHz, CDCl_3_) δ 206.10, 140.38, 130.20, 128.26, 125.69 (q, *J* = 3.7 Hz), 124.72, 122.92 (s,^1^*J* = 271.8), 55.31, 46.95, 45.22, 25.50. HRMS calcd. for C_13_H_19_FNO_3_S ([M + H]^+^): 288.1070, found 288.1075.

*Data for N-(4-chlorobenzyl)-3,3-dimethyl-2-oxobutane-1-sulfonamide* (**P-44**): Yield 23%; white solid; m.p. 104.8–108.0 °C. ^1^H-NMR (600 MHz, DMSO-*d*_6_) δ 7.73 (t, 1H, SO_2_-*N*H), 7.42 (d, 2H, C_6_H_4_-2H), 7.37 (d, 2H, C_6_H_4_-2H), 4.44 (s, 2H, CH_2_-SO_2_), 4.19 (d, 2H, *N*H-CH_2_), 1.08 (s, 9H, 3CH_3_). ^13^C-NMR (151 MHz, CDCl_3_) δ 206.11, 134.83, 133.91, 129.46, 128.87, 55.37, 46.83, 45.17, 25.53. HRMS calcd. for C_13_H_18_ClNO_3_NaS ([M + Na]^+^): 326.0594, found 326.0585.

*Data for N-(4-cyanobenzyl)-3,3-dimethyl-2-oxobutane-1-sulfonamide* (**P-45**): Yield 13%; white solid; m.p. 118.8–120.5 °C. ^1^H-NMR (600 MHz, DMSO-*d*_6_) δ 7.84 (dd, 3H, SO_2_-*N*H+ C_6_H_4_-2H), 7.55 (d, 2H, C_6_H_4_-2H), 4.52 (s, 2H, CH_2_-SO_2_), 4.30 (d, 2H, *N*H-CH_2_), 1.10 (s, 9H, 3CH_3_). ^13^C-NMR (151 MHz, CDCl_3_) δ 206.09, 141.99, 132.47, 128.43, 118.39, 111.81, 55.43, 46.84, 45.29, 25.51. HRMS calcd. for C_14_H_19_N_2_O_3_S ([M + H]^+^): 295.1116, found 295.1111.

*Data for N-(2,5-dimethoxybenzyl)-3,3-dimethyl-2-oxobutane-1-sulfonamide* (**P-46**): Yield 29%; white solid; m.p. 58.6–60.8 °C. ^1^H-NMR (600 MHz, DMSO-*d*_6_) δ 7.39 (t, 1H, SO_2_-*N*H), 6.96 (d, 1H, C6H4-H), 6.90 (d, 1H, C_6_H_4_-H), 6.81 (dd, 1H, C_6_H_4_-H), 4.40 (s, 2H, CH_2_-SO_2_), 4.18 (d, 2H, *N*H-CH_2_), 3.72 (d, 6H, 3OCH_3_), 1.06 (d, 9H, 3CH_3_). ^13^C-NMR (151 MHz, CDCl_3_) δ 205.45, 153.27, 151.60, 125.11, 116.01, 113.71, 111.09, 55.77 (d, *J* = 6.6 Hz), 55.53, 44.67, 44.39, 25.63. HRMS calcd. for C_15_H_23_NO_5_NaS ([M + Na]^+^): 352.1195, found 352.1193.

### 3.3. Fungicidal Activity Bioassays

The antifungal activity of the compounds was evaluated by a mycelial growth experiment, a tomato pot experiment, and a strawberry experiment.

*Mycelium Growth Experiments*. The method is given in reference [24]. The culture media was potato dextrose agar (PDA). The final concentrations of compounds was 50 mg/L on PDA, and each treatment was repeated three times. Boscalid was used as positive control while acetone was used as blank control. The pathogen cake (5.00mm) was inoculated in the center of PDA culture dish for 5 days. The inhibition rate I (%) was calculated according to following formula:I (%) = [(C − T)/(C − 5)] × 100%(1)

C is the average fungal mycelium diameters of the blank control (mm);

T is the average fungal mycelium diameters of the treatment (mm).

*Tomato Pot Experiment.* The method is given in reference [25]. The compound tested (200 mg/L) was evenly sprinkled on the tomato seedlings, and a spore suspension was inoculated 24 h later. Each treatment was repeated three times. When the rate of diseased leaves in the blank control reached more than 50%, the classification investigation was carried out. The inhibition rate I (%) was calculated according to the following formula:X = [∑(Ni × i)/(N × 9)] × 100%(2)
where X is the disease index, Ni the number of diseased leaves, i the disease grade and N the total leaves investigated.
I (%) = [(CX − TX)/(CX − 5)] × 100%(3)

CX is the average disease index of the blank control;

TX is the average disease index of the treatment.

*Strawberry Experiment.* The method is given in reference [25]. The compound to be tested (400 mg/L) was evenly sprinkled on the strawberries, and the spore suspension was inoculated 24 h later. Each treatment was repeated three times. When the disease spot diameter of the blank control was greater than 30 mm, the investigation wase carried out. The inhibition rate I (%) was calculated according to the following formula:I (%) = [(CM − TM)/CM] × 100%(4)

CM is the average lesion diameters of the blank control (mm);

TM is the average lesion of the treatment (mm).

## 4. Conclusions

In summary, 46 pinacolone sulfonamides were designed and synthesized, and presented excellent antifungal activity against a variety of plant pathogenic fungi, especially *Botrytis cinerea*. The high activity compound **P-29** was screened by biological activity evaluation and presented superb inhibition efficacy. Compound **P-29** has high research value and great potential to be developed as the novel lead compound in the design of a new pesticide.

## Data Availability

Not applicable.

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
