# Peer review of "Design, Synthesis and Structure-Activity Relationship of Novel Pinacolone Sulfonamide Derivatives against *Botrytis cinerea* as Potent Antifungal Agents"

_molecules, 2022, doi:10.3390/molecules27175468_

Round 1

Reviewer 1 Report

The manuscript requires major revisions.

 In Scheme 1, compounds P1 to P46 are not identified.

The section "3.1. Chemistry" must be entirely rewritten. Mainly, I am suggesting that an experienced synthetic organic chemist to be consulted. The experimental procedure described there does not make any sense !!!

Spectral data are wrong presented. For a 300 MHz proton NMR spectra the frequency of 13C cannot be 151 MHz. Mass spectrometry data are also wrong reported.

Typographical mistakes must be corrected.

Author Response

Point 1: In Scheme 1, compounds P1 to P46 are not identified.

Response: The R base in scheme 1 is shown in Table 1, and the structural identification is located in the "3.2. General Synthetic Procedures " part. Thank you very much for your suggestion.

Point 2: The section "3.1. Chemistry" must be entirely rewritten. Mainly, I am suggesting that an experienced synthetic organic chemist to be consulted. The experimental procedure described there does not make any sense !!!

Response: Thank you very much for your suggestion. This part has been modified in the paper, see “3.2. General Synthetic Procedures” in the revised manuscript, please.

Point 3: Spectral data are wrong presented. For a 300 MHz proton NMR spectra the frequency of 13C cannot be 151 MHz. Mass spectrometry data are also wrong reported.

Response: We are sorry for that we have made a mistake that the frequency of 1H NMR in this paper is 600 MHz, not 300 MHz, so the frequency of 13C NMR is correct. We have revised these data in the revised manuscript. The compounds with wrong MS data were identified again by HRMS instrument, and those wrong mass spectrometry data have also been corrected in the revised manuscript. See “3.2. General Synthetic Procedures” in the revised manuscript and “2.Mass spectrometry” in the revised supplementary material, please.

Point 4: Typographical mistakes must be corrected.

Response: Thank you very much for your suggestion. We have revised the manuscript according to the format requirements of the journal.

Reviewer 2 Report

The Authors presented synthesis of series of pinacolone sulfonamide derivatives followed by their preliminary antifungal activity investigations. The taken topic is important in the context of protection of fruit growing. This manuscript deserves publication in Molecules and dedicated issue on pesticides, however purities of the biologically evaluated compounds should be confirmed, e.g. with HPLC.

Author Response

Point 1: The Authors presented synthesis of series of pinacolone sulfonamide derivatives followed by their preliminary antifungal activity investigations. The taken topic is important in the context of protection of fruit growing. This manuscript deserves publication in Molecules and dedicated issue on pesticides, however purities of the biologically evaluated compounds should be confirmed, e.g. with HPLC.

Response: Thank you for your affirmation of this paper. We measured the purity of 5 biologically active compounds by HPLC analysis. According to the HPLC data, the purity of compounds P-18, P-23, P-29, P-30, P-31 were 97.1%, 98.9%, 87.7%, 95.0%, 99.7%. See “3.HPLC spectra” in the revised supplementary material, please.

Round 2

Reviewer 1 Report

The manuscript can be published in its actual form.